# Potential Combinatory Effect of Cannabidiol and Triclosan Incorporated into Sustained Release Delivery System against Oral Candidiasis

**DOI:** 10.3390/pharmaceutics14081624

**Published:** 2022-08-03

**Authors:** Mark Feldman, Irith Gati, Ronit Vogt Sionov, Sharonit Sahar-Helft, Michael Friedman, Doron Steinberg

**Affiliations:** 1Biofilm Research Laboratory, Institute of Dental Sciences, Faculty of Dental Medicine, The Hebrew University of Jerusalem, Jerusalem 91120, Israel; ronit.sionov@mail.huji.ac.il (R.V.S.); dorons@ekmd.huji.ac.il (D.S.); 2The Institute for Drug Research, Medical Faculty, The Hebrew University of Jerusalem, Jerusalem 91120, Israel; irith.gati@mail.huji.ac.il (I.G.); michaelf@ekmd.huji.ac.il (M.F.); 3Department of Endodontics, Faculty of Dental Medicine, The Hebrew University of Jerusalem, Jerusalem 91120, Israel; sharonit.sahar-helft@mail.huji.ac.il; 4Department of Endodontics, School of Graduate Dentistry, Rambam Health Care Center, Faculty of Medicine, Technion—Israel Institute of Technology, Haifa 31096, Israel

**Keywords:** *C. albicans*, CBD, triclosan, SRV, biofilm

## Abstract

*Candida albicans* is a common fungal pathogen. Biofilm formation on various surfaces is an important determinant of *C. albicans* pathogenicity. Our previous results demonstrated the high potential of cannabidiol (CBD) to affect *C. albicans* biofilms. Based on these data, we investigated the possibility of incorporating CBD and/or triclosan (an antimicrobial agent that is widely utilized in dentistry) in a sustained-release varnish (SRV) (SRV-CBD, SRV-triclosan) to increase their pharmaceutical potential against *C. albicans* biofilm, as well as that of the mixture of the agents into SRV (SRV-CBD/triclosan). The study was conducted in a plastic model, on agar, and in an ex vivo tooth model. Our results demonstrated strong antibiofilm activity of SRV-CBD and SRV-triclosan against *C. albicans* in all tested models. Both formulations were able to inhibit biofilm formation and to remove mature fungal biofilm. In addition, SRV-CBD and SRV-triclosan altered *C. albicans* morphology. Finally, we observed a dramatic enhancement of antibiofilm activity when combined SRV-CBD/triclosan was applied. In conclusion, we propose that incorporation of CBD or triclosan into SRV is an effective strategy to fight fungal biofilms. Importantly, the data demonstrate that our CBD/triclosan varnish is safe, and is not cytotoxic for normal mammalian cells. Furthermore, we propose that CBD and triclosan being in mixture in SRV exhibit complementary antibiofilm activity, and thus can be explored for further development as a potential treatment against fungal infections.

## 1. Introduction

*Candida albicans* is an opportunistic fungal pathogen, isolated from different sites in the human body, including the oral cavity. *Candida* is associated with high virulence in humans and often causes severe invasive mucosal and systemic infections, particularly in immunocompromised individuals. Oral candidiasis is linked to the formation of white plaques on inflamed mucosa in the oral cavity and is associated with immune suppression factors, such as smoking, dentures, corticosteroid use, diabetes mellitus, nutritional deficiencies, immunosuppressive diseases, hormone therapy, vitamin deficiencies, head and neck radiation, and uncontrolled usage of antibiotics [1,2]. One of the most important factors involved in *C. albicans* pathogenesis is attributed to biofilm formation on either biotic or abiotic surfaces [2,3,4,5]. There is a strong correlation between biofilm formation and *C. albicans* pathogenicity in oral candidiasis. Biofilm-associated pathogens are protected from disinfectants and antifungal agents. Moreover, throughout biofilm development, cells detach from biofilms, allowing Candida to disseminate to new areas, thus enhancing the pathogen’s virulence. Biofilm formation generally proceeds through three major stages: initial attachment to the surface, colonization, and maturation. The mature *C. albicans* biofilm represents a complex multilayer network of various cell types, including yeast-form cells, pseudohyphal cells, and hyphal cells, all embedded into the protective extracellular matrix [2,3,4,6]. 

To combat biofilm formation, various modalities are being sought. We hypothesized that cannabidiol (CBD) and/or triclosan may be particularly effective against oral *Candida* biofilm, especially when incorporated into a controlled-release delivery system.

Cannabidiol (CBD) is one of the major phytocannabinoids, produced by the *Cannabis sativa* plant. It has been shown to be non-psychoactive and well-tolerated in humans [7,8]. CBD has been demonstrated as a promising effective therapeutic agent in the treatment of various disorders including inflammatory and neurodegenerative diseases, autoimmune diseases, cardiovascular diseases, and cancer [9]. Several studies have reported a potential role for CBD as an antimicrobial agent [10,11,12,13,14,15]. CBD has been shown to have potent anti-bacterial activity against methicillin-resistant *Staphylococcus aureus* (MRSA) [12,16]. In combination with bacitracin, CBD reduced the minimal inhibitory concentration (MIC) value of this antibiotic against MRSA 64-fold [12]. Recently, we also reported antifungal activity of CBD against *C. albicans* [5]. 

Triclosan is a nonionic, broad spectrum, antimicrobial agent that has been incorporated into a variety of personal-care products. Triclosan is also widely used in dentistry. Triclosan toothpaste has shown high efficiency in keeping a healthy perio-implant environment [17,18] and in controlling periodontal conditions in children [19]. A resin composite matrix with incorporated triclosan demonstrated strong inhibition of three oral bacteria associated with dental caries: *Streptococcus mutans*, *Actinomyces viscosus,* and *Lactobacillus casei* [20].

The major disadvantage with drug-delivery systems for the oral cavity is their low residence time. Together with that, the low intrinsic substantivity of a vast majority of the common orally active substances leads to their inability to maintain effective therapeutic concentrations of the drug at the target site, which represents a major pharmacological disadvantage. Development of sustained-release delivery systems for local application may offer a promising strategy for overcoming such problems [21]. Polymer-based oral sustained-release varnishes (SRV) are examples of delivery systems in which the release of the active agent in the oral cavity is controlled. Diffusion and erosion are two major mechanisms of controlled drug release [22]. Matrix diffusion includes transfer of the drug molecules into the medium through the polymeric matrix of the SRV. Erosion represents chemical, biological, or physical degradation of the polymer surface or in bulk, followed by release of the entrapped drug. In order to further develop and optimize delivery systems, and to better understand the release mechanism, mathematical modelling is sometimes employed as a useful tool [23]. 

A challengingly low susceptibility of *C. albicans* biofilms to common antifungals represents a serious drawback for the treatment of biofilm-associated infections. In addition, many of the currently used antifungal drugs are highly toxic and might also interact with other drugs [24,25]. Moreover, the majority of antimycotic drugs exhibit short substantivity in the oral cavity in the form in which they are currently administrated, e.g., nystatin mouthwashes or lozenges. Therefore, there is an urgent need not only for alternative compounds capable of treating fungal biofilms with an improved safety profile, but also for enabling delivery systems of such agents. Several SRVs for prolonged release of drugs in the oral cavity have been developed for the treatment and prevention of oral candidiasis [6,21,26,27,28,29,30,31,32,33].

The aim of the present study was to investigate the possibility of incorporating CBD, triclosan, and CBD/triclosan into a sustained-release varnish (SRV-CBD, SRV-triclosan, SRV-CBD/triclosan) in order to increase the clinical potential of these agents against *C. albicans* biofilm. 

## 2. Materials and Methods

### 2.1. Materials

Active agents, CBD (Figure 1A) and triclosan (Figure 1B) were supplied by NC Laboratories (Prague, Czech Republic) and Merck (Rehovot, Israel), respectively. Polyethylene glycol 400 (PEG 400) were provided by Merck, Israel. Hydroxypropyl cellulose (Klucel™ EF grade was used) and aminomethacrylate copolymer (Eudragit™ E100) were supplied by Israeli representatives of Hercules™ and Evonik™, respectively. Ethanol was used as absolute, supplied in HPLC grade by J. T. Baker, Israel. 

### 2.2. Preparation of the SRV

The formulations were prepared according to the quantities of the materials as enumerated in the Table 1 below:

Briefly, PEG 400 was accurately weighed in a scintillation vial, equipped with a suitable magnetic stirring bar. Ethanol, 90% by weight of the formulation, was then added, and the polymers were added, consecutively, into vigorously mixed solvent. The mixture was maintained on hot plate set to 40 °C, until complete dissolution was achieved. The temperature was then brought to ambience, and the active ingredients were added, followed by mixing until dissolution. Thus a 10% wt. SRV was obtained. Thus, SRV-CBD and SRV-Triclosan contained about 2.2% wt. of the active compound, and SRV-CDB/Triclosan contained about 1.8% wt. of each active compound.

### 2.3. Dissolution Testing of SRV-CBD/Triclosan

To evaluate the release rate of the active ingredients from the varnish, the varnish was applied onto sterilized extracted human teeth (Helsinki approval 0406-17-HMO). Prior to the experiment, the teeth were kept in 70% ethanol for 24 h, then dried at 40 °C until constant weight was obtained. Each tooth was coated with the varnish, and dried at 40 °C for 1 h. The weight difference of the sample before and after coating, corresponded to the net weight of the applied coating.

The dissolution testing was conducted in a heat-room set to 37 °C. Each tooth was placed into 20 mL of phosphate buffer USP, pH = 6.8, fortified with 3% wt. of Tween 80, to ensure sink conditions for both cannabidiol and triclosan. The specimens were left on a shaker at 30 rpm, and samples were periodically extracted. The volume was replenished with a fresh aliquot of the heated medium. Samples were stored refrigerated until the analysis of the active agents. 

### 2.4. Quantification of the Active Ingredients

The quantification was performed using HPLC, as follows. CBD was quantified no later than 24 h after the sampling. The stability of the sample in the dissolution medium had been established prior to selection of the medium. The samples were analyzed using an Hitachi HPLC (Merck, Rehovot, Israel), an ACME C8 (3 µm) 150/4.6 mm HPLC column, equipped with a suitable pre-column (ACME 5/4.6 mm 3 µm), with UV detection at 211 nm in a temperature-controlled environment at 25 °C. An injection volume of 50 µL was used. CBD was eluted at 1 mL/min with the mobile phase consisting of 70% of acetonitrile by volume and 30% of 3 mM phosphate buffer at pH 3.0. 

Triclosan was quantified from the same samples, consecutively to cannabidiol, using an Hitachi HPLC (Merck, Rehovot, Israel) and a LichroCART RP-18 (5 µm) 150/4.6 mm HPLC column, equipped with a suitable pre-column (LichroCART 4-4 RP-18 (5 µm). UV detection was at 280 nm, in a temperature-controlled environment at 25 °C. An injection volume of 100 µL was used. Triclosan was eluted at 1 mL/min with the mobile phase consisting of 80% of acetonitrile by volume and 20% of 10 mM ammonium acetate solution.

### 2.5. Coating Persistence on the Teeth

A total of about 5 g of accurately weighed teeth and tooth fragments was used. The teeth were collectively dip-coated with the SRV, and dried at 40 °C, as described above. Several coats were applied, and the weight of the coating was determined gravimetrically. The teeth were placed in 20 mL of phosphate buffer 6.8 USP in a 50 mL centrifuge tube, and the tube was rotated using an automatic inverter in a heat-room (37 °C). The teeth were periodically removed, washed with double-distilled water to remove the residues of buffer, and dried to constant weight. 

### 2.6. Fungal Strains and Growth Conditions

Two fungal strains, *C. albicans* SC5314 and *C. albicans* SC5314 carrying GFP reporter gene (*C. albicans*–GFP) [34] supplied by Professor J. Berman (Tel Aviv University, Tel Aviv, Israel), were grown for several days at room temperature (RT) on potato dextrose agar (PDA) plates (Neogen, Lansing, MI, USA). The fungi were resuspended at OD_600_ = 0.05 in RPMI medium (Sigma-Aldrich, St. Louis, MO, USA) and used for further assays. 

### 2.7. The Effect of SRVs on Biofilm Formation on Plastic Surfaces

Different volumes of SRV-placebo (control), SRV-CBD, SRV-triclosan and SRV-CBD/triclosan were applied into a 6-well microplate and dried overnight at 37 °C to form SRV membranes under sterile conditions. An overnight-grown culture of *C. albicans* was diluted at OD_595_ = 0.05 in RPMI medium and added at different volumes to the wells with SRVs. The final concentrations of each tested agent or combination in SRV were 100 µg and 200 µg per ml of fungal inoculum in the well. Biofilms were grown for 24 h at 37 °C in RPMI medium. The planktonic cells’ viability in the supernatant fluid of the formed biofilms was tested by MTT assay [5].

The amounts of *Candida* biofilms formed in the wells, including on the SRVs were assayed as follows: after washing the biofilms three times with PBS to remove loosely adhering cells, the metabolic activity of the *Candida* cells immobilized in the biofilms was analyzed by MTT assay [5], while total biomass was measured using crystal-violet staining [35]. The results are presented as the percentages of biofilm formation in samples treated with different SRVs compared to the biofilm formation in controls (SRV-placebo, 100%). Assays were performed in triplicate.

### 2.8. The Effect of SRVs on Preformed Biofilms on Plastic Surfaces

To investigate the effect of SRVs on preformed biofilms, *C. albicans* biofilms were allowed to mature for 24 h at 37 °C, as described above. The biofilms were washed twice with PBS. Dried aliquots of 12.5 μL and 25 μL of SRVs (dried membranes) of placebo, CBD, triclosan or CBD/triclosan mixture were applied into the wells with the pre-formed biofilms. Fresh RPMI medium was added to the wells, giving final concentrations for the agents alone and their combination of about 100 µg/mL and 200 µg/mL (corresponding to 12.5 μL and 25 μL of SRV, respectively). The plates were further incubated for 24 h at 37 °C. The amounts of remaining *C. albicans* biofilm were determined quantitatively using a standard MTT assay [5]. The assays were performed in triplicate.

### 2.9. The Effect of SRVs on Biofilm Formation and Hyphae Formation on Agar

The assay was performed on YPD agar supplemented with glucose 1% in 12-well plates. For hyphae initiation and formation, agar was supplemented with fetal calf serum (FCS) 10%.

Briefly, SRVs of placebo, CBD, triclosan or CBD/triclosan mixture were poured on agar giving final concentrations for the agents alone and their combination of about 100 µg per ml of agar, with following drying at RT and SRV membrane development. For hyphae formation, an aliquot of 8 µL of *C. albicans* at OD_595_ = 0.05 in RPMI medium was placed in the middle of each dried SRV and the plates were incubated at 37 °C for 14 days. 

For biofilm formation, 100 µL of *C. albicans* at OD_595_ = 0.05 in RPMI medium was applied on each dried SRV membrane on the agar in the well and plates were incubated at 37 °C, for 14 days.

Biofilm and hyphae formation were observed visually each day of incubation and photographed after 14 days. The assays were performed in triplicate.

### 2.10. Ex Vivo Tooth Model

The teeth were coated with SRV-placebo, SRV-CBD, SRV-triclosan and SRV-CBD/triclosan, as described above, for the dissolution testing. Three individual coats were applied. The SRV-coated teeth were incubated in 12-well plates with *C. albicans*–GFP at OD_595_ = 0.05 for 24 h at 37 °C in RPMI medium. Following washing with PBS, the biofilms formed on SRV teeth were visualized using a Nikon confocal microscope (Nikon Inc. Melville, NY) at excitation and emission wavelengths of 488 and 522 nm, respectively. At least three random fields were captured for each sample. The number of fungal cells in each sample was calculated according to the fluorescence intensity using Image J v3.91 software (http://rsb.info.nih.gov/ij, accessed on 2 August 2022). Data are presented as percentage of the control (SRV-placebo).

### 2.11. Cytotoxicity Assay

African green monkey normal Vero kidney cells kindly provided by Dr. Alex Rouvinsky (The Hadassah Medical School—The Hebrew University of Jerusalem, Israel) were cultivated in DMEM (Sigma) supplemented with 8% heat-inactivated newborn-calf serum (FCS) (Sigma). For measuring cytotoxicity, 4 × 10^4^ Vero cells were seeded per well in 200 µL of DMEM (Sigma) supplemented with 8% FCS and incubated at 37 °C in a humidified atmosphere of 5% CO_2_/95% air. The following day, 200 µL of the medium incubated with the SRV-placebo or SRV-CBD/triclosan varnishes was added directly to the Vero cells. After a 24 h incubation, the cell morphology was inspected under an inverted light microscope, and the metabolic activity measured by the MTT assay. For the MTT assay, 50 µL of a 5 mg/mL MTT (Sigma) solution was added to each well of 200 µL fluid. After a 45 min incubation at 37 °C, the supernatants were removed and the formazan formed was dissolved in 100 µL dimethylsulfoxide (DMSO). The absorbance was measured at 570 nm using the Tecan Infinite M200 plate reader. 

12.5 µL or 25 µL of SRV-placebo or SRV-CBD/triclosan was used to coat 24-well tissue-culture-grade plates corresponding to 200 and 400 µg of active agent. After film formation, 2 mL of DMEM supplemented with 8% FCS was added to the coated wells, and the plate was incubated at 37 °C overnight to allow the release of the active compounds to the medium. The medium was then collected for the cytotoxicity assay described above.

### 2.12. Statistical Analysis

Means of three independent experiments ± standard errors of the means (SEM) were calculated. The statistical analysis was performed using Student’s *t*-test with a significance level of *p* < 0.05 as compared to controls.

## 3. Results

### 3.1. The Drug Release from SRV-CBD/Triclosan

A graph of the cumulative release profile as a function of time is presented in Figure 2. At the dissolution conditions used, i.e., at sink conditions, the drug release for both active agents followed very closely the Higuchi release model. The dissolution profiles of the SRV-CBD and SRV-triclosan demonstrated a very similar release profile, further corroborating the basic model assumption (not shown).

### 3.2. The Coating Resilience

The aim of the experiment was to evaluate the resilience of the coating to grinding stresses applied by the hard surfaces of the teeth, which occur in a variety of daily activities, including mastication. No significant change in the coated teeth’s weight was observed during the first 5 h of the experiment. However, at 24 h, the final weight was lower than the initial weight of the teeth. Washing the teeth in ethanol to remove the residue revealed that no coating remained after 24 h of continuous “teeth grinding” in the inverter, indicating that the teeth indeed experienced weight loss after 24 h, under the experimental conditions. 

### 3.3. Inhibitory Effect on Biofilm Formation in Plastic Model

The results concerning the SRV model clearly showed an inhibitory effect of all tested agents’ formulations on fungal biofilm formation as compared to SRV-placebo. This effect was dose-dependent. SRV-CBD exhibited a more pronounced inhibitory effect then SRV-triclosan especially in reduction of viable cells in the biofilm. SRV-CBD at 100 µg/mL and 200 µg/mL decreased viable cells by 43% and 68%, respectively, while SRV-triclosan at the above tested doses was able to reduce viable cells by 23% and 50%, respectively, as compared to SRV-placebo (Figure 3). In addition, at 200 µg/mL, SRV-CBD and SRV-triclosan inhibited biofilm biomass by 64% and 50%, respectively (Figure 3). However, the most pronounced inhibitory effect was observed when both agents were incorporated together into SRV as a mixture. Both viable cells and total biomass were reduced by more than 70% as compared to SRV-placebo, when the combination of CBD and triclosan in SRV was applied at 100 µg/mL (Figure 3). Finally, SRV-CBD/triclosan at 200 µg/mL was able to reduce, almost completely, the viable cells and total biomass of the *C. albicans* biofilm (Figure 3). Interestingly, SRV membrane-associated biofilm was formed on SRV-triclosan only, while it was not detected on other tested formulations (Figure 3). Very importantly, the viability rates of planktonic fungi in the experiment were not affected to any significant extent by any of tested agents’ formulations (data not shown).

### 3.4. Eradication of Matured Biofilm in the Plastic Model

In addition to notable inhibition of biofilm formation, the SRV of the tested compounds and their combinations also dramatically removed pre-formed fungal biofilm in a dose-dependent manner. As for biofilm inhibition, SRV-CBD was also more effective in mature biofilm detachment than SRV-triclosan. When SRVs were applied at 100 µg/mL of each agent alone or in combination, SRV-CBD and SRV-triclosan removed matured biofilm by 40% and 20%, respectively, as compared to SRV-placebo (Figure 4). Increasing the dose to 200 µg/mL caused further eradication of pre-formed biofilm by SRV-CBD (63%) and SRV-triclosan (52%) as compared to SRV-placebo (Figure 4). Finally, the combination SRV-CBD/triclosan at 100 µg/mL drastically removed matured biofilm by 80% as compared to SRV-placebo, significantly enhancing antibiofilm activity of each tested agent alone (Figure 4), while 200 µg/mL of the mixture of the agents in SRV significantly removed pre-formed biofilm as compared to each tested agent alone (Figure 4).

### 3.5. Long-Term Inhibition of Biofilm Formation and Fungal Morphology Alteration on Agar

Long-term inhibition of biofilm formation and fungal morphology alteration on agar is shown in Figure 5. As demonstrated in Figure 5A, after 14 days of incubation on agar with SRV-placebo, hyphae initiation and development from *C. albicans* colonies were clearly visible. The image represents two morphological phenotypes of fungi *C. albicans*: a smooth yeast colony and filamentous hyphae with the image showing the transition phase from yeast to hyphae. 

In contrast, both SRV-CBD and SRV-triclosan prevented formation of fungal filaments, while still allowing colony growth. Interestingly, in addition to inhibition of hyphae development, SRV-CBD also prevented the colony spreading within the agar. Finally, the combination SRV-CBD/Triclosan totally inhibited growth of *C. albicans* colonies.

Similarly, pronounced long-term (14-day) inhibition of biofilm formation on SRV-CBD/triclosan membrane was observed. *C. albicans* was able to form a biofilm around the SRV-CBD/triclosan membrane and on its edges, however almost no colony growth was observed in the middle of the membrane (Figure 5B). SRV-CBD was less effective than SRV-CBD/triclosan, allowing greater membrane covering by fungal biofilm (Figure 5B). Finally, SRV-triclosan and SRV-placebo had no effect on *C. albicans* biofilm formation, since strong and intact biofilm was formed on their membranes and around the membranes (Figure 5B).

### 3.6. Biofilm Inhibition in Ex Vivo Tooth Model

In addition to the notable anti-biofilm effect on the plastic model, the tested formulations also successfully inhibited fungal biofilm formation on human teeth. Similarly, to the plastic model, SRV-CBD was more potent than SRV-triclosan in reducing biofilm formation in ex vivo by 75% (Figure 6B,E), while SRV-triclosan inhibited biofilm formation by 50% (Figure 6C,E) as compared to SRV-placebo (Figure 6A,E). SRV of CBD and triclosan in combination was even more effective than SRV of each compound applied alone. SRV-CBD/triclosan demonstrated a drastic inhibition of biofilm formation by almost 90% (Figure 6D,E) as compared to SRV-placebo (Figure 6A,E).

### 3.7. SRV-CBD/Triclosan Does Not Exhibit Cytotoxicity towards Mammalian Cells

The cytotoxic effect of CBD/triclosan released from the varnishes was tested on Vero cells which are considered the gold standard for this purpose. Microscopic inspection of the Vero cells following the 24 h incubation with medium that had been incubated with the placebo or CBD/triclosan varnishes, showed normal morphology, indicating that the CBD/triclosan released from the varnishes was not cytotoxic to the Vero cells. The metabolic activity of the cells exposed to the CBD/triclosan medium was similar to that of cells exposed to the placebo medium (Figure 7). These data demonstrate that our CBD/triclosan varnish is safe, and is not cytotoxic for normal cells. 

## 4. Discussion

Due to the numerous disadvantages of commonly used antimycotics, there is an urgent need for alternative antifungal agents. Attachment and accumulation on living and abiotic surfaces is the major virulence determinant of the pathogenic fungi. Denture stomatitis, the most common form of oral candidiasis affects the majority of the elderly population wearing denture prosthesis. *Candida* biofilm formation and progression occurring on the prosthesis–mucosa interface leads to mucosal inflammation. Prosthesis-associated biofilm and inflammation can cause severe pain, difficulty in eating and speaking, and device malfunction, thus negatively impacting the lifestyle of the patient [36]. Therefore, a novel alternative approach should be focused on alteration of fungal biofilm formation capabilities [37,38]. For a long time, common drugs, such as amphotericin B, clotrimazole, flucytosine, miconazole, nystatin, itraconazole, and ketoconazole have been widely used as antifungal treatment. However, numerous disadvantages including side effects, high risk of resistant strain emergence, and low therapeutic potential limited their clinical use. Conventional topical antifungals are sometimes applied in the form of oral gels, lozenges, or varnishes, and are aimed at treating uncomplicated and localized candidiasis, such as denture stomatitis [39,40,41]. In addition, low substantivity of the drug in the oral cavity may compromise the efficacy of the active agent. This major pharmacological disadvantage could be overcome by incorporating the active agent into a sustained-release delivery system. The pharmaceutical technology of sustained-release delivery systems of antibacterial agents has been extensively reported for the treatment of oral bacterial infections such as dental caries and periodontitis [21,26,28,42,43,44,45,46]. However, limited scientific reports are documented regarding the prolonged utilization of such lasting formulations against oral fungal diseases [6,29]. 

Previously, we have reported that controlled release of triclosan from the sustained-release delivery system extended its anti-biofilm properties against *S. mutans* [21]. Our current study demonstrated a strong inhibitory effect of SRV-triclosan and SRV-CBD on *C. albicans* biofilm formation in plastic and in ex vivo tooth models. In addition, in an agar model, both formulations demonstrated a long-term inhibitory effect on hyphae development of the *Candida*, which is considered a major pathogenic pathway of *Candida* (i.e., the transition from yeast to hyphae). The filaments in fungal biofilms support the development of stable and intact structure. Hyphae facilitate tissue invasion, damage, and escape from host cells leading to pathology and potentially death. Previously it was reported that invasive candidiasis affects more than 250,000 people worldwide every year and it is estimated to cause over 50,000 deaths annually [47]. Furthermore, due to low immune protection against oral candidiasis in the elderly population, the virulence factors of *Candida* such as filamentation, tissue invasion, and associated inflammation in prosthesis users are critically important.

Therefore, altering the morphological form could be a potential strategy in the fight against *C. albicans* infection. This effect of the agents released from the formulated SRV is supported by data reported in the literature concerning the anti-filamentation activity of triclosan and CBD in solution [5,48]. Previously we reported that CBD exhibits anti-*Candida* activity through a multi-target mode of action including downregulation of the expression of virulence genes and alteration of cell and mitochondrial membranes and cell-wall function [5]. 

Moreover, both formulations were able to partially remove a pre-formed fungal biofilm in a dose-dependent manner. While all tested formulations effectively impaired mature biofilm at 100 µg/mL and 200 µg/mL, none of the varnishes affected fungal viability and growth at these concentrations. Similarly, our previous findings demonstrated specific antibiofilm non-fungicidal activity of SRV formulation applied against *C. albicans* [6]. In contrast, most of the conventional antifungal drugs can only affect mature biofilms at concentrations that are much higher than their effective dose for planktonic cells, indicating a high resistance rate of *C. albicans* biofilms to antifungals [49,50]. 

Although the single agent formulations had a minor-to-moderate antibiofilm activity, SRV-CBD and SRV-triclosan, combinations of the two tested compounds incorporated into an SRV, had a significantly more pronounced antibiofilm effect in all the tested models. The combinatory effect of these agents incorporated into an SRV may indicate that the two compounds affect biofilm formation in an independent manner, yet the mechanisms whereby they affect their changes on the biofilm formation are likely to be related. The effect could be attributed to the cumulative release of the agents from the varnish leading to elevated biological activity of SRV-CBD/triclosan, but the effects are seen in dose-dependent manner for each compound, which further supports the unexpected combinatory effect between the two modalities. Dramatic inhibition of biofilm formation and total removal of pre-formed biofilm was detected in the presence of SRV-CBD/triclosan in the plastic model. Mature biofilm removal by SRV-CBD/triclosan is extremely critical, since *C. albicans* biofilms are often associated with medical implants, enhancing disease the severity and mortality rate of the patients [51]. 

Furthermore, SRV-CBD/triclosan exhibited prolonged antibiofilm effect on agar. Notably, this formulation was able to inhibit the fungal colony growth and biofilm formation on the sustained-release membrane for 14 days. We propose that gradual and constant release of CBD and/or triclosan from the SRV leads to a cumulative inhibitory effect and supports the prolonged activity of the formulation. The long-term presence of the agents in the vicinity of the biofilm maintained the concentration gradients needed for effective diffusion, and allowed for their better penetration into the deep layers of the biofilm, reaching inhibiting or fungicidal concentration of the drug in the deeper layers of the biofilm. This finding is in line with previously reported data that the active agents incorporated into polymeric SRV membrane are capable of exhibiting activity against oral candidiasis in a prolonged manner [6,29,30,31,52]. 

Interestingly, in contrast to the drastic inhibition of biofilm formation on and around the SRV-CBD/triclosan membrane in the plastic model, biofilm formation around the membrane on agar was not affected. This could be attributed to different patterns of release kinetics of active agents from the SRV in various media such as liquid medium and agar. It is of note that sink conditions are not necessarily expected to be maintained in vivo, and the drug release may be significantly lower when fluid restriction is present, or when fluid turnover is low. The permeation of clotrimazole loaded into nanocapsules through the mucosa and receptor medium have been studied in non-sink conditions. This formulation was considered promising and suitable for vaginal application against candida-related infections [53].

The results obtained from the ex vivo tooth model clearly showed a pronounced inhibitory effect of SRV-CBD/triclosan on biofilm formation on the human tooth, seconding the finding on the plastic substrate. In parallel, as detected by the coating-resilience assay, the coating was capable of withstanding, at least initially, the friction of the teeth on one another, and of performing under conditions that eventually ground down the teeth, something that is not normally expected to occur during mastication in the oral cavity. The results also indicate that the coating will eventually peel off from the teeth and will not pose any health risk. Yet, the clinical efficacy of the combination SRV-CBD/triclosan will have to be demonstrated in properly controlled clinical trials.

As for the pharmaceutical formulations of the SRV, both drugs were released at a similar rate, almost synchronously in fact, from their varnishes, and from their combined varnish. The drug release closely followed the diffusion-based drug-release model, known as the Higuchi model [54,55]. The amount released up to 70% of the drug loading (one of the model’s limitations), which is directly proportional to the square root of time, with the coefficients of determination of above 0.98 (data not shown). Given that both polymers used in the varnish are insoluble at the dissolution conditions of temperature and pH values of the medium, the basic assumptions for the model, i.e., the non-degradable matrix and planar geometry, were maintained, and it can therefore be concluded that the both drugs were released by diffusion through the controlled-release matrix. Quite intriguingly, the terminal parts of the graphs also follow the same release model, but with a significantly lower diffusion coefficient, consistent with a possible porosity being developed in the film over time and thus an increased diffusion path and the viscosity of the medium, due to presence of the hydrogel in the formulation.

Different technologies of drug delivery have been suggested for the oral cavity [41,42,43]. Clearly, each of these pharmaceutical applications has its pros and cons. The main disadvantage of most of the oral-delivery applications is the low substantivity of the active agents in the oral environment, which affects their clinical performance. Since dental diseases are of a chronic nature, long-term pharmaceutical treatment is often necessary. Therefore, the duration of the active agent in the oral cavity is an important parameter in the prevention and treatment of dental-associated diseases. SRVs are suitable applications for such a modality. Application of any dental medications should be easy to perform by nonprofessional as well as professional people. Indeed, the SRV can be easily applied at home by properly educated patients. In addition, as the drug is released over time, the effect of patient compliance is minimized. However, it should be mentioned that oral sustained-release deliveries systems such as SRVs should be regarded only as supplementary to the traditional daily oral hygiene routine of tooth brushing and mouth rinses, to ensure protection between these very important treatments. 

Despite numerous studies investigating the antimicrobial activity of single agents incorporated into SRVs, there is very limited information concerning the combinatory effect of agent mixtures introduced into SRVs, in part due to the complexity of the data analysis involved. The pharmaceutical idea of using a combination of antibiofilm agents such as CBD and a traditional antimicrobial agent, incorporated into a sustained-release delivery system such as an SRV, to the best of our knowledge, has never been suggested, at least not in dental medicine. Our developed formulation of an SRV containing a mixture of non-classic fungicidal compounds exerting an antibiofilm effect, which fundamentally differs from common antifungals, represents a promising novel concept and strategy for the treatment of oral fungal diseases. Our results have demonstrated, for the first time, the high potential of a combination of non-fungicidal agents, such as CBD and triclosan, incorporated into an SRV against *C. albicans* biofilm, as a useful anti-biofilm modality.

## Figures and Tables

**Figure 1 pharmaceutics-14-01624-f001:**
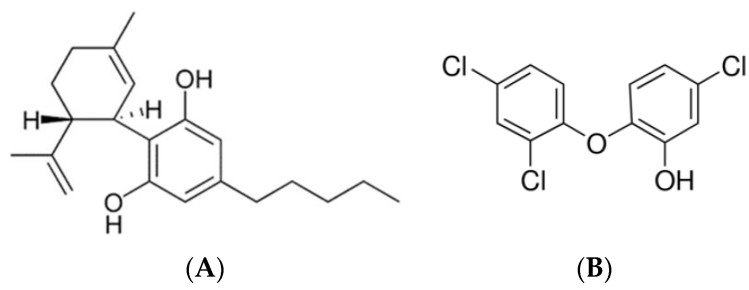
The structure of CBD (**A**) and triclosan (**B**).

**Figure 2 pharmaceutics-14-01624-f002:**
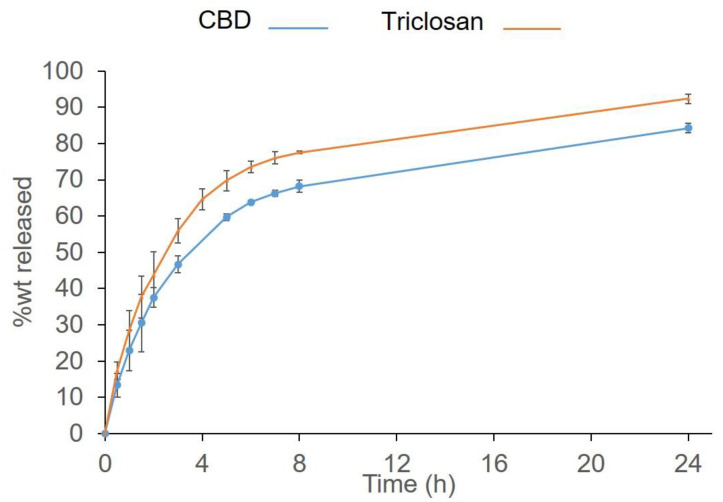
The cumulative release of the drugs under sink conditions.

**Figure 3 pharmaceutics-14-01624-f003:**
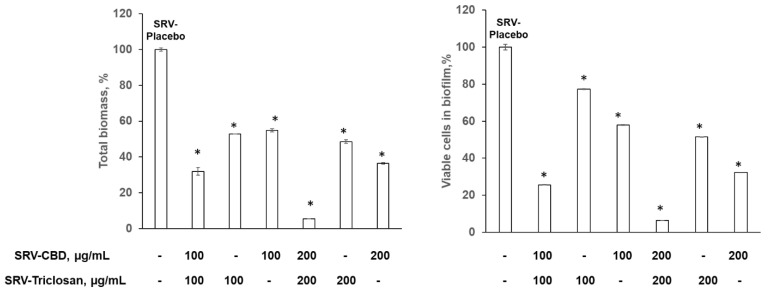
Inhibitory effect on biofilm formation in plastic model. Fungal biofilms were formed in the presence of 100 µg/mL of SRV-CBD, SRV-triclosan, and SRV-CBD/triclosan; quantitative analysis of total biomass and viable cells in biofilm. A value of 100% was assigned to *C. albicans* biofilms treated with SRV-placebo. * Significantly lower than the value for the control (*p* < 0.05). SEM from three independent experiments were calculated.

**Figure 4 pharmaceutics-14-01624-f004:**
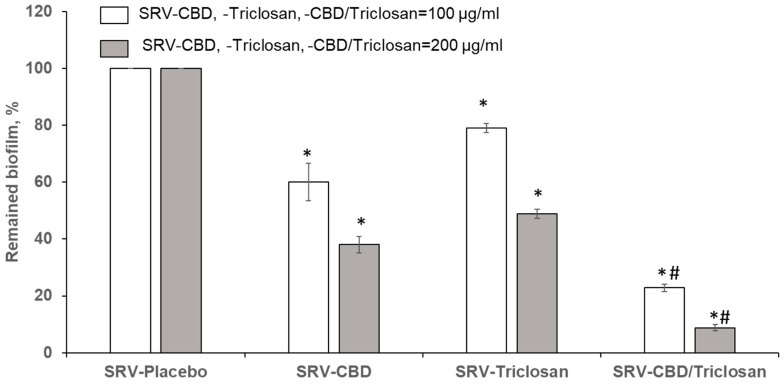
Eradication of matured biofilm in the plastic model. Quantitative analysis of viable cells in remaining biofilms. A value of 100% was assigned to *C. albicans* biofilms treated with SRV-placebo. * Significantly lower than the value for the control (*p* < 0.05). ^#^ Significantly lower than the value for each tested agent alone (*p* < 0.05). SEM from three independent experiments were calculated.

**Figure 5 pharmaceutics-14-01624-f005:**
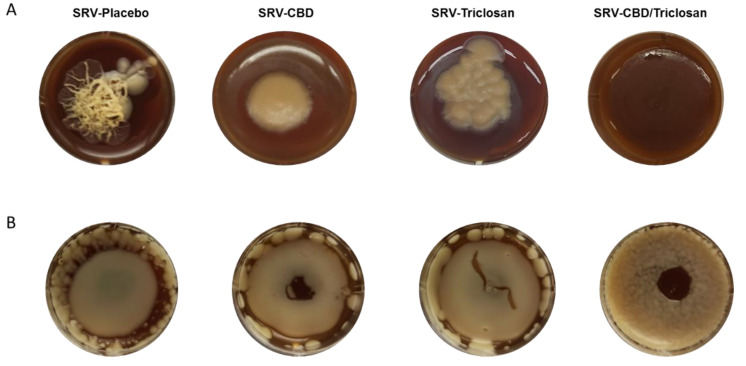
Prolonged inhibitory effect on agar. (**A**) *C. albicans* hyphae initiation and development on agar for 14 days with the presence of SRV formulations. (**B**) Fungal biofilm formation for 14 days on agar with the presence of SRV formulations. SRV-placebo served as control. Assay was performed in triplicate.

**Figure 6 pharmaceutics-14-01624-f006:**
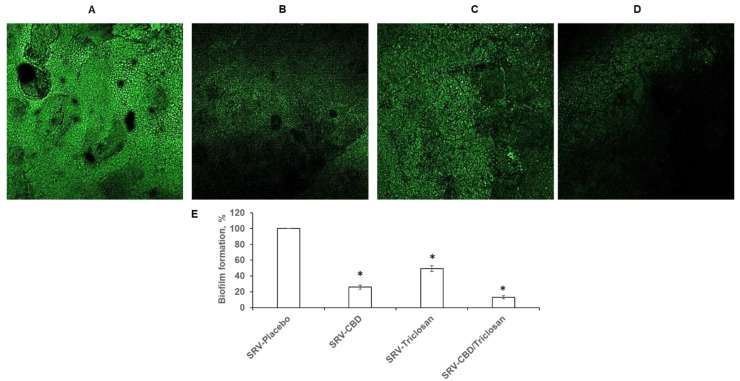
Biofilm inhibition in ex vivo tooth model. *C. albicans*–GFP biofilm formation on teeth with presence of SRV formulations: (**A**) SRV-placebo, (**B**) SRV-CBD, (**C**) SRV-triclosan, and (**D**) SRV-CBD/triclosan. Magnification: 40×. Image processing was performed using Nikon confocal microscope at excitation and emission wavelengths of 488 and 522 nm, respectively. (**E**) Quantitative analysis of fluorescence images. The biofilm amount in each sample was calculated according to the fluorescence intensity using Image J v3.91 software (http://rsb.info.nih.gov/ij, accessed on 2 August 2022). A value of 100% was assigned to *C. albicans*–GFP biofilm treated with SRV-placebo. * Significantly lower than the value for the control (SRV-placebo) (*p* < 0.05). At least three random fields were observed and analyzed. Assays were performed in triplicate and the SEM from three independent experiments were calculated.

**Figure 7 pharmaceutics-14-01624-f007:**
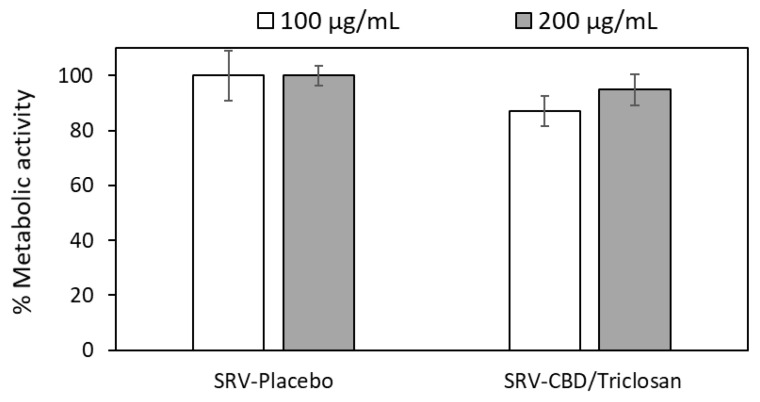
The CBD/triclosan released from the varnish is not cytotoxic to Vero cells. SRV-placebo and SRV-CBD/triclosan varnishes containing 200 and 400 µg active compounds were incubated in 2 mL of medium overnight (represented as 100 µg/mL (white bars) and 200 µg/mL (grey bars), respectively). Then, 200 µL of each supernatant was incubated on pre-seeded Vero cells for 24 h. The metabolic activity of the CBD/triclosan-exposed cells was measured by MTT assay and compared to the placebo-exposed cells. *n* = 9 for each treatment group. There was no significant difference between the groups.

**Table 1 pharmaceutics-14-01624-t001:** The qualitative composition of the SRV.

Materials	SRV-CBD/Triclosan	SRV-CBD	SRV-Triclosan	SRV-Placebo
CBD	18.09%	22.09%	-	-
Triclosan	18.09%	-	22.09%	-
Klucel^®^ EF	18.29%	22.33%	22.33%	28.66%
PEG 400	9.04%	11.04%	11.04%	14.17%
Eudragit^®^ E	36.48%	44.54%	44.54%	57.17%

Generally the SRV formulation was conducted similarly to previous studies [30,31].

## Data Availability

Raw data available upon reasonable request

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
