# Peer review of "Potential Combinatory Effect of Cannabidiol and Triclosan Incorporated into Sustained Release Delivery System against Oral Candidiasis"

_pharmaceutics, 2022, doi:10.3390/pharmaceutics14081624_

Round 1

Reviewer 1 Report

The authors discribes a Potential combinatory effect of cannabidiol and triclosan incorporated into sustained release delivery system against oral candidiasis.

The study was well conducted and is aggregates important information for the scientific Community

Line 43:  Please insert the correlation of biofilms in the pathogenesis of oral candidiasis

Line 99: What MIC values were obtained for cannabidiol that were used as the basis for the design of the proposed formulation?

Line 149: What cell concentration is this optical density corresponding to?

Line 155: The biofilm formation assay is poorly described. Where is the description of the adhesion phase and the growth phase?

Line 165: This time is not usual to evaluate the maturation of Candida biofilm. The biofilm may still be immature

Line 171: Why was violet crystal not used to assess the amount of biomass eliminated?

Line 176: How were the compounds spread evenly on the plate ensuring this concentration?

Line 179: Was the sample sowed only in YPD agar to compare the colonies?

Figure 2: A third line exemplifying the uncontrolled release time would make the data much better to be assimilated

Figure 4: These images look very similar. Visually there is no difference between the wells

Figure 5 – PLACEBO - This image is very confusing, apparently two phenotypes in the same image, or a contamination.

Figure 6 - There is no way to identify what is being shown in the image

Line 297: Describe about the formation of biofilm in oral candidiasis, mainly reinforcing the users of prostheses

Line 314: Italics

Line 315: Describe that in addition to the importance in biofilms, filamentation in Candida is associated with tissue invasion.

Line 335: highlight this importance in prosthesis users

Author Response

Dear Editor,

We thank the editorial board and the reviewers for their valuable remarks.

We have modified the manuscript according to their comments. (All corrections throughout the manuscript are track–changed).

We hope now that the manuscript will be accepted to your journal.

Dr. Mark Feldman

on behalf on the co-authors.

Reviewer 2 Report

The manuscript is well written and shows interesting data on a promising future therapeutic possibility for dental biofilm control and fungal infection control.

The methods have been carefully and carefully detailed. 

Perhaps the only suggestion to the authors is that they consider incorporating a brief paragraph in the discussion of a possible mechanism of anti-candida action of cannabinoids. 

Author Response

(The authors gave the same response as above.)

Reviewer 3 Report

The article presented by Felman et al. shows an interesting work that demonstrates a strong antifungal activity of cannabidiol/triclosan sustained release varnish (SRV) against C. albicans. This SRV-cannabidiol/Triclosan showed a sustained release that was able to inhibit biofilm formation.

This article shows an interesting scientific work, and the discussion is supported by the results obtained. The work is clear and well written. However, we consider that for its publication in this journal it is necessary to include some in vitro study that completes the scientific level of this work.

To improve this work, we propose the following changes:

Major revision

MTT Cytotoxicity assay.

Cytotoxicity studies of CBD, Triclosan and CBD/Triclosan SRV should be evaluated using an MTT assay. The concentrations that do not show cytotoxicity should be related to the concentrations released from the SRV systems.

You could select one or two cell lines that allow you to evaluate the different concentrations of CBD, Triclosan and SRV-CBD/Triclosan using different culture times between 2-24h. Cell surviving capability could be estimated by MTT assay.

Discussion.

Line 349. “It is of note that the sink conditions are not necessarily expected to be maintained in vivo, and the drug release may be significantly lower when fluid restriction is present or when fluid turnover is low”.

We consider that it would be appropriate to include different antifungals that have been studied in no sink conditions and include their references.

Minor revision

-Material.

-lines 90-94. For all active agents and excipients, you must include the city before the country

-line 99. Table 1. The table does not include lines in the row of columns.

-line 127. You must include the HPLC column model in parentheses

-line 132. you must include the HPLC pre-column model in parentheses

-line 138. Change the symbol to 5 g. (not 5 grams).

-line 149. Quitar el segundo punto final.

-line 175. The full stop in this text is missing.

-line 220. Figure 3. Eliminate the upper panels A and B (they do not provide new information). Their results are clearly seen in the figures for Total biomass, % and Viable cells in biofilm, %.

-line 252. Figure 4. Delete the upper panel A (100 and 200 µg/ml). You could include it, perhaps, as supplementary material. This panel does not provide new information. In this figure it is necessary to include the error bar for SRV-CBD/Triclosan (100 and 200 µg/ml).-Line 245. Delete the text (upper panel).

-Line 248. It is missing to include the remove matured biofilm percentage of SRV-CBD/Triclosan indicating its significant difference (p<0.05) compared to treatments with CBD or Triclosan alone for 100 µg/ml.

-Line 249. Delete the text (upper panel).

-Line 249. The phrase “While 200 µg/ml of mixture of the agents in SRV almost totally eradicated”… This expression is not correct. Figure 4 shows a value between 5-10%. You must indicate the remove matured biofilm percentage of SRV-CBD/Triclosan indicating if it presents a significant difference compared to treatments with CBD or Triclosan alone for 200 µg/ml.

-Line 254. Delete the text (upper panel).

References.

It is necessary to adjust the references to the standards of the journal.

1. Author 1, A.B.; Author 2, C.D. Title of the article. Abbreviated Journal Name YearVolume, page range.

Line 403. El año es en negrita, el volumen sin negrita. Es importante incluir doi.

Eg.

1. Mayer, F., D. Wilson, and B. Hube, Candida albicans pathogenicity mechanisms. Virulence 4: 119-128. 2016.

to be replaced by

1. Mayer, F., Wilson, D., and Hube B. Candida albicans pathogenicity mechanisms. Virulence 2013, 4: 119-128. https://doi.org/10.4161/viru.22913

Author Response

(The authors gave the same response as above.)

Reviewer 4 Report

Thanks for this interesting paper.

I would like to congratulate the authors on this nice paper. 

1. The introduction is satisfactory, which justified the onset of this research work, but I would like to raise the authors' attention that the Study design is missing.

2. How many teeth were analyzed?

Author Response

(The authors gave the same response as above.)

Round 2

Reviewer 3 Report

We consider that the answers to the considerations presented are appropriate and that the changes made in the work presented by Felman et al., improve the scientific interest of this paper and make it suitable for publication in Pharmaceutics.